# Removal and Reclamation of Trace Metals from Copper and Gold Mine Tailing Leachates Using an Alkali Suspension Method

**Shunfeng Jiang** [1,2,*] **, Yali Chen** [3] **, Siqin Chen** [3] **and Ziying Hu** [3]

1   College of Life and Environmental Science, Wenzhou University, Wenzhou 325035, China
2   National and Local Joint Engineering Research Center for Ecological Treatment Technology of Urban Water Pollution, Wenzhou University, Wenzhou 325035, China
3   CAS Key Laboratory of Urban Pollutant Conversion, Department of Applied Chemistry, University of Science and Technology of China, Hefei 230026, China
*   Correspondence: sfjiang@wzu.edu.cn

**Abstract:** Leachates from mine tailing ponds, which usually contain a variety of metallic ions, are highly toxic to human and ecological health. The common methods used to remove those trace metallic are difficult due to the extremely acidic conditions and the diverse kinds of metals in mine tailing leachates. Herein, we proposed an economical and efficient soil-assisted alkali suspension approach to remove and reclaim the trace metals. Under the optimum conditions, more than 98% of $Cu^{2+}$, $Zn^{2+}$, and $Cr^{3+}$, and 93% of $Cd^{2+}$ were removed from authentic copper and gold tailing leachates. Multiple characterizations indicated that the quick removal of trace metal ions from leachates was mainly due to the formation of amorphous hydroxides which are easily adsorbed by soil particles. Additionally, small quantities of metal ions and organic matter complexes were formed, which contributed to the removal of trace metals. Furthermore, most of the adsorbed trace metal in authentic tailing leachate can be reclaimed by a simple acid treatment. Life cycle assessment analysis demonstrated the environmental sustainability of this alkali suspension method due to its smaller contribution to global warming. This study provides an efficient and low-cost approach for the disposal and recycling of toxic mine tailing leachates.

**Keywords:** mining tailings; leachate; alkali soil suspension; trace metals; reclaim





## 1. Introduction

The amount of mine tailings has drastically increased with the development of the mining industry, and most tailings are stored in ponds because they still contain some valuable noble metals and toxic trace metals [1–3]. Due to their long-term storage, leachates containing ions and compounds with many toxic trace metals (such as As, Fe, Cu, Zn, Cd, Co, Ni, Pb, Hg, Tl, Se, Te, Cr, and Sb) are inevitably produced, and can pose a severe threat to human and ecological health [4–6]. This threat is even more pronounced when considering the possible catastrophic failure of tailings ponds [7–9].

Great efforts have been made to prevent leachate permeation. For instance, Ahn et al. used a solidified cover system (an engineered hardpan) to reduce water infiltration, acid generation, and sulfide oxidation [10]. Shen et al. reported that tailings pond anti-seepage engineering technology can effectively reduce environmental pollution [11]. However, even though construction quality is strictly controlled, leaching is still unavoidable. In particular, the possibility of a dam break is extremely dangerous to downstream ecosystems. Thus, an economical and efficient method is urgently desired for the treatment and subsequent recovery of valuable metals from tailing wastewater.

Various physicochemical and biological processes have been applied for the treatment of mine tailing leachates. Biological systems including anaerobic biological reactors and

microbial mats, which can be used to remove and immobilize the trace metal in leachates through microbial activity [12]. However, these biological processes are susceptible to various factors including organic carbon sources, leachate properties, bacterial diversity and activities, and reactor configuration and conditions [13,14]. The commonly used physicochemical methods for removing trace metal ions from polluted aqueous solutions are adsorption, ion exchange, pressure-driven membrane separation, permeable reactive barriers, and photocatalysis [15,16]. Permeable reactive barriers constructed with a buried barrier that intercepts the flow of groundwater represent the most commonly used physicochemical technology for the treatment of mine water due to their minimum needs in terms of control, supervision and maintenance [17]. For example, a permeable reactive barrier using zero-valent iron was employed to quickly remove reducible metals (e.g., $Cr^{6+}$ and $Cu^{2+}$) [18,19]. A drawback of this method is the accumulation of precipitates (mainly hydroxides, other iron corrosion products, and carbonates salts) that clog the membrane pores, which in turn decreases its hydraulic permeability and pollutant removal efficiency [18]. Adsorption techniques are generally regarded as the most effective and convenient method for removing trace metal ions from polluted aqueous solutions. However, the use of common adsorbents, such as activated carbon, aluminium oxide, chitosan, etc., is quite unaffordable for the treatment of a mass of mine tailing leachates [20]. Recently, Zhao et al., developed an $Au/g\text{-}C_3N_4/Co_3O_4$ heterojunction catalyst for the photocatalysis removal of $Cr^{6+}$, and found that 85.6% of $Cr^{6+}$ can be removed via the photocatalysis system [21]. In general, the high energy cost, materials, and need for human operation of these technologies have caused researchers to search for cost effective and environmentally friendly technologies to treat mine tailing leachates.

Considering the diversity of trace metals (i.e., reductive and oxidative) in mine tailings, chemical precipitation by adjusting the solution pH is a broad-spectrum method for the simultaneous removal of different toxic metals [22–24]. Consequently, novel reagents and materials have been studied to separate trace metal ions by chemical precipitation. Ye et al. extracted more than 99% of the zinc from lead-zinc mine tailings by using $Na_2S$ [25]. Zeng et al. used ball milling activated $CaCO_3$ to authenticate a lead recovery rate of 99% from lead-zinc solution [26]. However, chemical precipitation displays low efficiency for authentic leachates that contain diverse metal ions with different concentrations because the newly formed metallic hydroxide particles are very small and have difficulties precipitating quickly. Additionally, the optimum pH for chemical precipitation was greater than 9.5; thus, large amounts of alkali reagents are required for the treatment of mine tailing leachates [27]. Adding solid materials to mine tailing leachates to accelerate the settlement of metallic hydroxide particles may be a feasible strategy for enhancing the chemical precipitation efficiency of trace metals. Fly ash is the most widely used additive for chemical precipitation due to its specific chemical and physical properties and low cost [28–30]. Considering that the composition of soil is similar to that of fly ash (silica, ferric oxide, calcium oxide, magnesium oxide, and carbon, glassy phase), the greater availability and lower transportation cost of soil make it an alternative for the treatment of mine tailing leachates.

Soil is no doubt the least expensive and most readily available additive for this purpose, and therefore soil-assisted chemical precipitation may be a promising method to quickly retain a variety of trace metals from tailing leachates. However, many factors that are closely related to the overall efficiency and cost of a soil-assisted treatment system need to be experimentally determined. For example, suspension pH is directly related to the amount of alkali needed, hydraulic retention time (HRT) is the primary design parameter when determining volumetric capacity, and temperature reflects the effect of seasonal variations. In addition, the interaction between trace metal ions and soil, and the effects of the inorganic and organic matter in the soil on the removal of trace metals need to be clarified.

In this study, a soil-assisted alkali suspension was used to remove different trace metal ions from two authentic mine tailing leachates (Figure S1). The optimum treatment

parameters (e.g., pH, HRT, and temperature) were determined based on batch experiments using synthetic leachates containing typical trace metals ($Cu^{2+}$, $Zn^{2+}$, $Cd^{2+}$, and $Cr^{3+}$). The effects of the inorganic and organic matter in the soil on the removal efficiency of trace metals were investigated. Thereafter, the mechanism of trace metal removal was discussed based on the Fourier transform infrared spectroscopy (FTIR), X-ray diffraction (XRD), and X–ray photoelectron spectroscopy (XPS) analyses.

## 2. Materials and Methods

### 2.1. Materials

The reagents used in this work were all analytical grade, purchased from Sinopharm Chemical Reagent Co., Ltd., Shanghai, China, and used without further purification. Copper, zinc, cadmium, and chromium solutions at a concentration of 50 mg $L^{-1}$ were prepared by dissolving an appropriate amount of cupric nitrate trihydrate ($Cu(NO_3)_2 \cdot 3H_2O$), zinc sulfate heptahydrate ($ZnSO_4 \cdot 7H_2O$), cadmium chloride hemipentahydrate ($CdCl_2 \cdot 2.5H_2O$), and chromium nitrate nonahydrate ($Cr(NO_3)_3 \cdot 9H_2O$) in distilled water in volumetric flasks.

The soil used as an absorbent was collected from a random location in one of the gardens located at the University of Science and Technology of China, Hefei, China. The soil was air-dried at room temperature, and then crushed by a high-speed rotary cutting mill. Soil particles with sizes between 0.15 mm and 0.30 mm were collected and stored in a plastic sealed pouch for subsequent experiments. The copper tailings and gold tailings were collected from a copper mine and gold mine in Fushun, Liaoning Province, China. Authentic leachates were obtained by leaching the tailings with acidic solutions for 25 days.

### 2.2. Determination of Trace Metal Concentrations

In the synthetic leachate experiments, the concentrations of copper, zinc, and cadmium were determined via inductively coupled plasma-atomic emission spectrometry (ICP-AES, Optima 7300 DV, Perkin-Elmer Co., Waltham, MA, USA) and the concentration of chromium was determined via UV-vis spectrophotometry (UV-1700, Phenix Co., Ltd., Shangrao, China) using a colorimetric method [31]. For the authentic mine tailing leachates experiments, the concentrations of the trace metals in the solutions after the removal experiments were analysed via ICP-AES or inductively coupled plasma-mass spectrometry (ICP-MS; used for relatively low concentrations of trace metal ions).

The point of zero charge ($pH_{PZC}$) of soil was determined following the method described by Saha et al. using an automated titrator (DL50 Mettler-Toledo) [32]. FTIR (EQUIVOX55 IR spectrometer, Bruker, Germany) with a resolution of 2 $cm^{-1}$ in the range of 4000 to 400 $cm^{-1}$ with 16 scans was used to characterize the functional groups on the surface of the raw soil and soil after sorption. XPS (ESCALAB250, Thermo-VGScientific Inc., Essex, UK) using monochromatized Al K$\alpha$ radiation (1486.9 eV) was used to analyse the compositions and valence state changes of C1s and N1s in the raw soil and soil after sorption. XRD (MXPAHF, Japanese Make Co., Tokyo, Japan) utilizing a nickel-filtered Cu K$\alpha$ radiation source (30 kV/160 mA, $\lambda$ = 1.54056 Å) scanned from 20° to 80° was applied to capture the differences in solid phase morphology of raw soil and soil after sorption.

### 2.3. Synthetic Leachate Experiment

Trace metal removal kinetics tests were performed using synthetic leachates containing a selected trace metal at pH 8.5 for different times and temperatures (10, 20 and 30 °C). A 50 mL aliquot of the single-metal (copper, zinc, cadmium, or chromium) solution was mixed with 1.0 g of soil and shaken in a thermostatic oscillator (WHY-2, China) with an oscillation frequency of 180 rpm. The initial solution pH (8.5) was adjusted by adding HCl or NaOH solutions and monitored with a pH meter (PHS-25, China). After different contact times (5, 15, 30, 60 min), a small portion of the suspension was sampled by filtration through a membrane filter (0.45 μm), and the trace metal (copper, zinc, chromium, and cadmium) concentrations were determined as described.

The multi-metal solution, which contained four trace metals (copper, zinc, cadmium, and chromium) each at a concentration of 50 mg L$^{-1}$, was prepared by dissolving a desired amount of the four metal salts mentioned above in the required amount of distilled water in a single volumetric flask. The raw soil (particle sizes between 0.15 mm and 0.30 mm) was incinerated in a muffle furnace at 600 °C for 6 h with a heating rate of 5 °C min$^{-1}$ to obtain organic-free soil. This experimental process was similar to the single-metal solution procedure. The experiment that removed the trace metals from the multi-metal solution was similar to that from the single-metal solution.

### 2.4. Authentic Leachate Experiment

The authentic copper tailing leachate (CuTL) and gold tailing leachate (AuTL) were obtained by mixing 1000 g of gold tailings in a 2000 mL 1 M $H_2SO_4$ solution, and by mixing 700 g copper tailings in a 1400 mL 1 M $H_2SO_4$ solution in two 5 L beakers, respectively. A 1 M $H_2SO_4$ solution was added to shorten the leaching time and extract more trace metals. The two beakers were sealed with plastic wrap and allowed to sit for 25 days. Then, the two suspensions were centrifuged to obtain the authentic leachates. The concentrations of copper, zinc, cadmium, and chromium in the authentic leachates were determined and shown in Table S1. Trace metal removal experiments, which were carried out with authentic leachates, were similar to those performed with the synthetic leachates.

A reclaim experiment was also conducted. The polluted soil suspension was filtered, dried at 60 °C overnight and weighed. Four grams of the dried soil was desorbed with 10 mL 1 M $H_2SO_4$ in 25 mL conical flask for 1 h in an oscillator with a frequency of 180 rpm and then filtered. The concentrations of $Cu^{2+}$, $Zn^{2+}$, $Cd^{2+}$, and $Cr^{3+}$ in the filtered $H_2SO_4$ solution were determined by atomic absorption spectroscopy (AAS).

### 2.5. Life Cycle Impact Assessment

The environmental impact and energy consumption of the proposed alkali suspension method for tailing leachates remediation were determined using the life cycle assessment (LCA) method. Based on principle of LCA, the scope of the inventory included pH conversion of wastewater, adsorption and separation of soil, landfill disposal of recycled soil and waste soil, background production of any chemicals, and the energy required for the treatment process. Because this study focused on the disposal process of tailing leachates, the environmental impacts resulting from the construction of this method were not considered [33]. A functional unit of 1 m$^3$ of tailing leachates was used under these different scenarios (Figure S2). Important category groups were used as a baseline in the LCA method.

## 3. Results and Discussion

### 3.1. Effect of pH

Most trace metals form precipitates in basic solutions. However, the optimal pH for the precipitation of different trace metals varies significantly. According to the species of trace metal and their solubility product ($K_{sp}$), the effects of pH on the relative distribution of $Cu^{2+}$, $Zn^{2+}$, $Cd^{2+}$, and $Cr^{3+}$ species (i.e., $M^{n+}$, $M(OH)^{(n-1)+}$, $M(OH)_2^{(n-2)+}$, etc.) in aqueous solutions were calculated using Visual MINEQ software, and the results are shown in Figure 1. The pH values at which the concentrations of $Cu^{2+}$, $Zn^{2+}$, $Cd^{2+}$, and $Cr^{3+}$ reached their minimum were 8.0, 10.0, 11.5, and 6.0, respectively. Clearly, the operational cost of treatment increased at elevated suspension pH. In a previous study [31], the addition of inexpensive adsorbents (e.g., sediment) during trace metal precipitation significantly improved trace metal removal efficiency by enhancing the adsorption and coprecipitation effects. The removal efficiency of trace metals increased when the soil content of the suspension was increased. Moreover, considering that alkali consumption would increase by approximately one order of magnitude when the pH increases from 8.5 to 10.0, a pH of 8.5 was adopted as the economically optimized pH value for the treatment of tailing leachates.

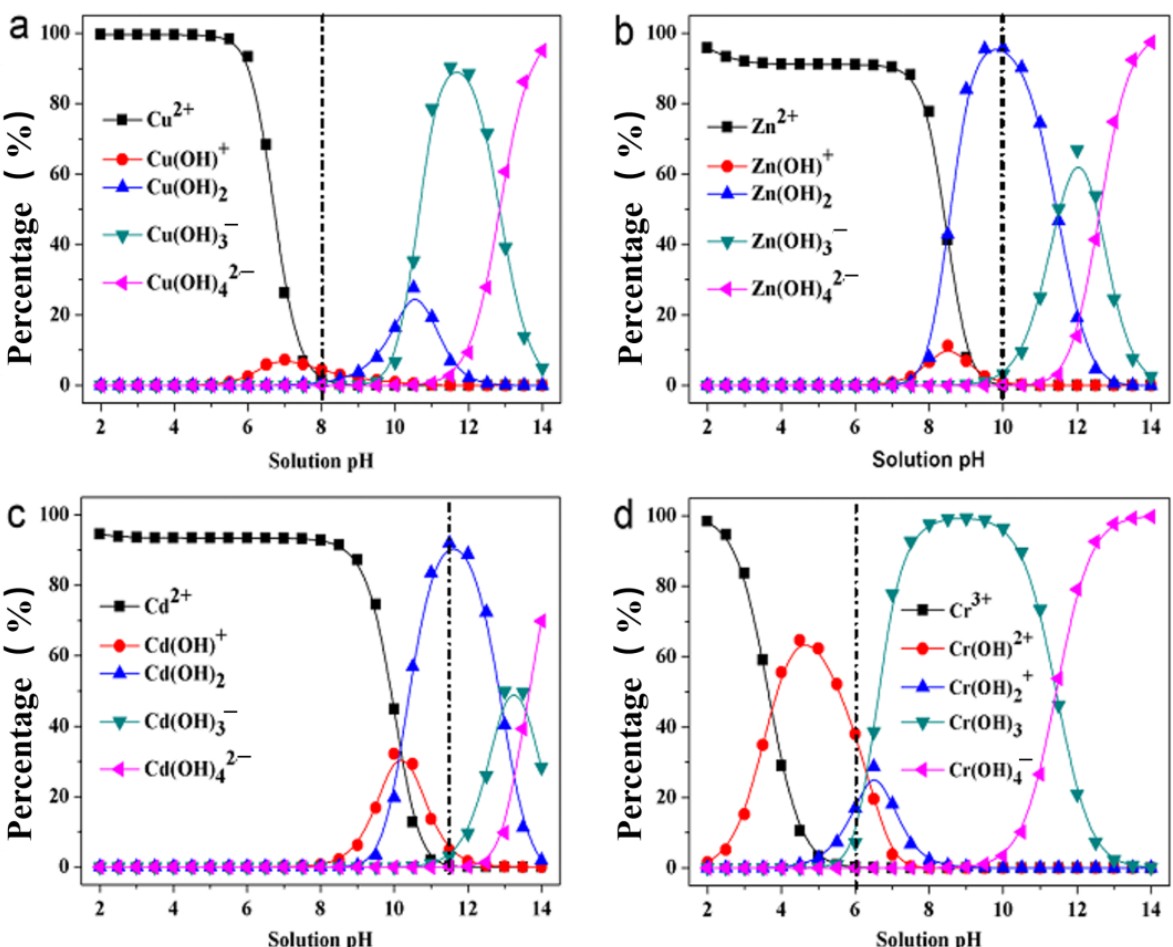

**Figure 1.** Speciation distribution of (**a**) Cu, (**b**) Zn, (**c**) Cd, and (**d**) Cr in aqueous solution at different pH.

### 3.2. The Appropriate HRT and Amount of Soil Additive

Figure S4 shows the effect of contact time on the removal efficiencies of $Cu^{2+}$, $Zn^{2+}$, $Cd^{2+}$, and $Cr^{3+}$ by the weakly alkaline soil suspension. For copper and zinc, it is clear that adsorption equilibrium was achieved within approximately 5 min and their removal efficiencies were almost 100%, while only approximately 80% of the $Cd^{2+}$ and $Cr^{3+}$ were removed in the first 5 min. Adsorption equilibrium of $Cd^{2+}$ and $Cr^{3+}$ was achieved in approximately 30 min. The removal of trace metals is mainly related to the precipitation and adsorption processes [34]. The theoretical equilibrium concentrations of different trace metals at a pH of 8.5 were calculated based on their $K_{sp}$ values (Equations (1) and (2), Table S1).

$$M(OH)_n \rightarrow M^{n+} + nHO^- \tag{1}$$

$$K_{sp} = [M^{n+}][HO^-]^n \tag{2}$$

The calculated results in Table S1 show that there is no obvious relationship between removal efficiency and equilibrium concentration of these trace metals. Thus, the formation of hydroxides is not the rate-controlling step in an alkali–soil medium, as this reaction is quick in this environment [35]. However, the newly formed fine grains of metal hydroxides are difficult to precipitate, which suggests that their adsorption by soil particles is the rate-limiting process [36]. A short contact time between the alkali–soil suspension and trace metals is desirable since this would reduce facility installation and operational costs. Therefore, the HRT was limited to 5 min according to Figure 2.

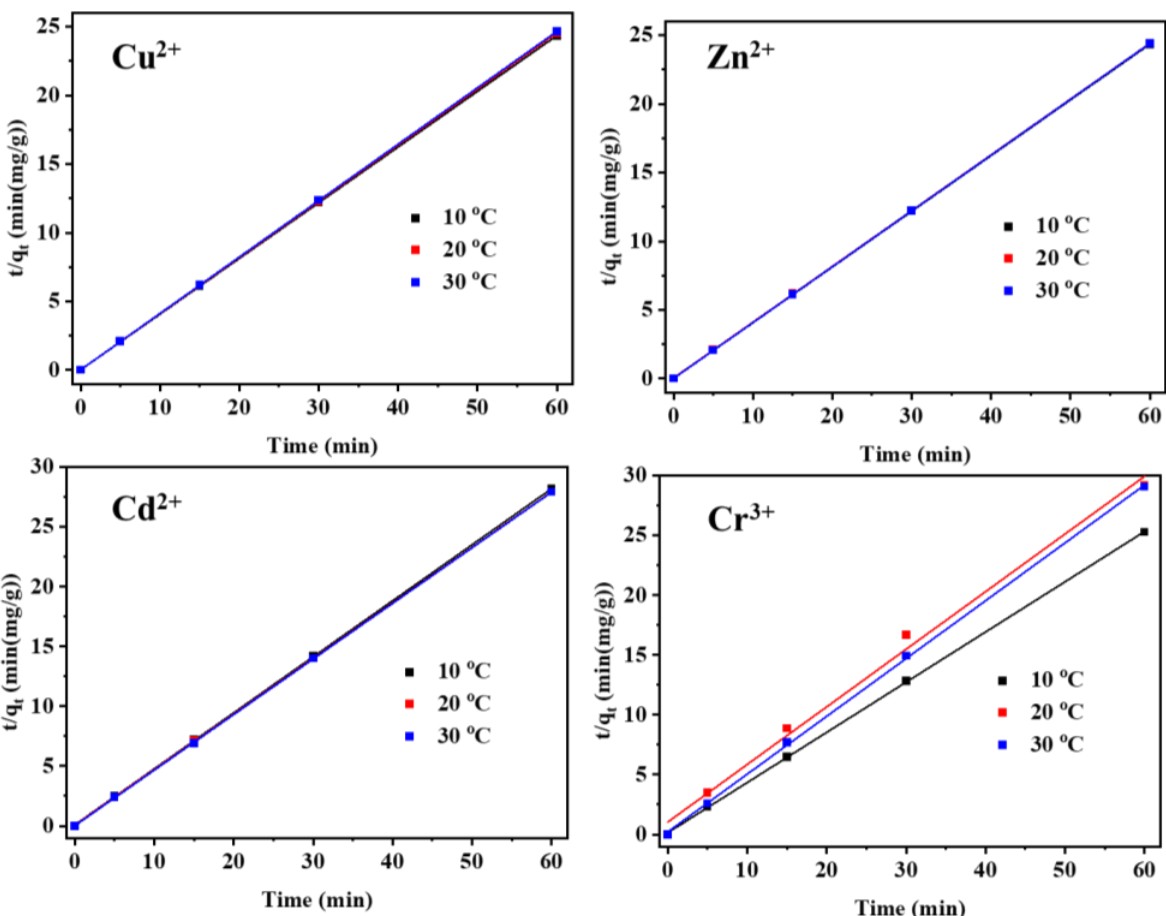

**Figure 2.** The pseudo-second-order kinetics model of trace metal removal in the alkali–soil system.

Pseudo-first- and pseudo-second-order kinetics models were employed to depict the adsorption process in the alkali–soil medium (Equations (3) and (4)) [37].

$$q_t = q_e \left(1 - e^{-k_1 t}\right) \tag{3}$$

$$\frac{t}{q_t} = \frac{1}{k_2 q_e^2} + \frac{t}{q_e} \tag{4}$$

where $q_t$ and $q_e$ are the amounts of trace metal adsorbed (mg g$^{-1}$) on the soil at time t (min) and equilibrium, respectively, and $k_1$ and $k_2$ represent the pseudo-first-order rate constant (min$^{-1}$) and pseudo-second-order rate constant (g·mg$^{-1}$·min$^{-1}$)), respectively.

The kinetic models of the different trace metals are shown in Figure 2 and Figure S5, and the fitting data are listed in Table 1. According to the coefficients of determination ($R^2$ values), the pseudo-second order kinetic model is more suitable to describe the removal process from the alkali–soil suspension. The parameters of the pseudo-second-order model ($q_e$ and $K_2$) were calculated from the intercept and slope of the $t/q_t$ versus t plot. The adsorption capacities ($q_e$ values, calculated) of the soil were in the range of 2.1–2.5 mg g$^{-1}$ for 50 mg L$^{-1}$ of each trace metal, which are close to the experimental values (2.5 mg g$^{-1}$) when adsorption equilibrium was reached. Moreover, these values were all in agreement with the activated sorption mechanism, indicating that chemisorption (chemical reaction) was the rate-controlling step of the adsorption process [38,39]. Thus, 20 g L$^{-1}$ soil and an HRT of 5 min are suitable for authentic copper or gold tailing leachates based on kinetic experiments.

**Table 1.** Fitting data of pseudo-first-order and pseudo-second-order kinetics model.

| Trace Metal | Temp. (°C) | Pseudo-First Order Kinetics | | | Pseudo-Second Order Kinetics | | |
|---|---|---|---|---|---|---|---|
| | | $q_e$ | $k_1$ | $R^2$ | $q_e$ | $k_2$ | $R^2$ |
| $Cu^{2+}$ | 10 | 2.46 | 0.92 | 0.9999 | 2.47 | 8.79 | 0.9999 |
| | 20 | 2.44 | 1.02 | 0.9999 | 2.45 | 10.52 | 0.9999 |
| | 30 | 2.43 | 1.07 | 0.9999 | 2.44 | 7.46 | 0.9999 |
| $Zn^{2+}$ | 10 | 2.46 | 0.92 | 0.9999 | 2.47 | 8.79 | 0.9999 |
| | 20 | 2.44 | 0.83 | 0.9998 | 2.46 | 3.31 | 0.9999 |
| | 30 | 2.46 | 0.92 | 0.9999 | 2.46 | 8.78 | 0.9999 |
| $Cd^{2+}$ | 10 | 2.12 | 0.57 | 0.9998 | 2.14 | 2.26 | 0.9999 |
| | 20 | 2.13 | 0.72 | 0.9994 | 2.16 | 2.74 | 0.9999 |
| | 30 | 2.16 | 0.63 | 0.9998 | 2.15 | 12.56 | 0.9999 |
| $Cr^{3+}$ | 10 | 2.35 | 0.47 | 0.9996 | 2.38 | 1.18 | 0.9999 |
| | 20 | 1.87 | 0.29 | 0.9687 | 2.08 | 0.22 | 0.9918 |
| | 30 | 2.01 | 0.69 | 0.9975 | 2.07 | 1.10 | 0.9998 |

*3.3. Trace Metal Removal Efficiency from Authentic Leachates*

To test the efficiency of soil suspensions for treatment of authentic tailings leachate under the economically optimized conditions, experiments were conducted using gold and copper mine tailings. As shown in Table S1, the initial concentrations of Cu, Zn, Cd, and Cr in the authentic tailing leachates were 8.53, 8.66, 0.40, and 43.58 mg L$^{-1}$ in Au-TL and 140.5, 138.9, 1.90, and 6.25 mg L$^{-1}$ in Cu-TL, respectively. The removal efficiencies of the trace metals in the authentic tailing leachates are shown in Figure 3a,b. After treatment with raw soil at a pH of 8.5, the final concentrations of Cu, Zn, and Cr were 0.10, 0.043, and 0.12 mg L$^{-1}$ at 10 °C in the copper tailings, respectively. The removal efficiency slightly decreased with an increase in temperature, and the final concentrations of Cu, Zn, and Cr were 0.42, 0.051, and 0.12 mg L$^{-1}$ at 30 °C, respectively. Cd was not detected in the treated solutions. Similar to the single-metal experiment, the removal efficiency of Cr at pH = 8.5 was slightly lower, which might be attributed to the repulsion between the negatively charged soil and Cr(III) anions at the experimental pH. For the AuTL, the removal efficiencies of all trace metals reached 100% with the exception of Cd at 30 °C, which was attributed to the poor precipitation performance and lower initial concentration of Cd$^{2+}$. Nevertheless, the concentrations of all tested trace metals in the treated solution were lower than the national primary discharge standards (GB 8978-1996) (total Cu 5.0, Zn 2.0, Cd 0.1, and Cr of 1.5 mg L$^{-1}$), which indicates that this alkali–soil suspension method can be used to treat tailing leachates.

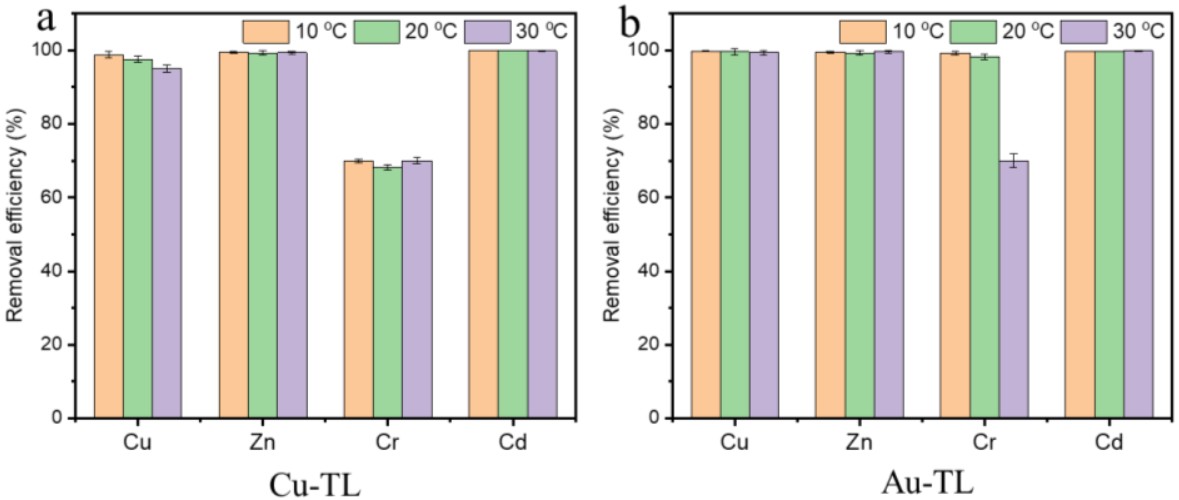

**Figure 3.** Removal efficiency of Cu, Zn, Cd and Cr in Cu-TL (**a**) and Au-TL (**b**) by soil.

### 3.4. Effect of Temperature

Mining leachates are usually treated outdoors, and seasonal influences on efficiency should be considered. Temperatures of 10, 20, and 30 °C were chosen to represent the conditions in winter, spring and autumn, and summer, respectively. The results of the experiments performed at different temperatures showed no significant differences in trace metal removal efficiency, with the exception of $Cr^{3+}$. For $Cr^{3+}$ removal, lower temperatures are favourable, which may be attributed to the exothermic reaction of Cr adsorption.

### 3.5. Effect of Soil Composition

Soil mainly consists of clay minerals, metal hydroxides, and soil organic matter. The removal mechanism of trace metals by soil has been reported to involve three important processes: adsorption, surface precipitation, and fixation [40]. To elucidate the effects of the different components in soil, trace metal removal experiments were carried out with synthetic leachates and different types of soil. The results of the experiments are shown in Figure S6, and the fitting to the pseudo-second-order kinetics model and corresponding parameters are shown in Table 2 and Figure 4. The removal efficiencies of all four investigated trace metals using raw, untreated soil were the highest (almost 100% within the initial 5 min), and those with the organic-free soil had relatively lower removal efficiencies, whereas the experiments without soil had the lowest removal efficiencies. Although most trace metals can form metal-hydroxide particles (Figure 1 and Table S1) in alkali solution, the fine particles are not easily removed [41,42]. Soil can trap these fine hydroxide particles and remove them from solution [43]. The experimental results also showed that natural organic matter (NOM) played some role in the removal of trace metals, which may be attributed to the effect of the adsorptive sites supplied by organic matter and complexation between soil organic matter and trace metals. In addition, sandy soil with fewer clay minerals and metal hydroxides than raw soil was also used to remove trace metals in alkali solution. According to the kinetics results, the trace metal removal performance of sandy soil in alkali solution was inferior to that of raw soil and organic-free soil.

**Table 2.** The pseudo-second-order kinetics parameters of trace metal fixation by different substrate.

| Substrate | Trace Metal | $q_e$ (mg/g) | $k_2$ (min/(mg/g)) | $R^2$ |
|---|---|---|---|---|
| Raw soil | $Cu^{2+}$ | 2.47 | 8.80 | 0.9999 |
| | $Zn^{2+}$ | 2.47 | 8.79 | 0.9999 |
| | $Cd^{2+}$ | 2.13 | 2.26 | 0.9999 |
| | $Cr^{3+}$ | 2.38 | 1.18 | 0.9998 |
| Organic-free soil | $Cu^{2+}$ | 2.34 | 9.30 | 0.9988 |
| | $Zn^{2+}$ | 2.31 | 3.25 | 0.9988 |
| | $Cd^{2+}$ | 2.11 | 1.74 | 0.9997 |
| | $Cr^{3+}$ | 2.29 | 0.96 | 0.9984 |
| Sandy soil | $Cu^{2+}$ | 1.50 | 8.17 | 0.9887 |
| | $Zn^{2+}$ | 1.98 | 1.17 | 0.9995 |
| | $Cd^{2+}$ | 1.53 | 1.53 | 0.9977 |
| | $Cr^{3+}$ | 1.87 | 2.96 | 0.9998 |
| Without soil | $Cu^{2+}$ | 0.39 | 1.06 | 0.6347 |
| | $Zn^{2+}$ | 0.37 | 1.39 | 0.6796 |
| | $Cd^{2+}$ | 0.23 | 1.10 | 0.4712 |
| | $Cr^{3+}$ | 0.30 | 0.55 | 0.8699 |

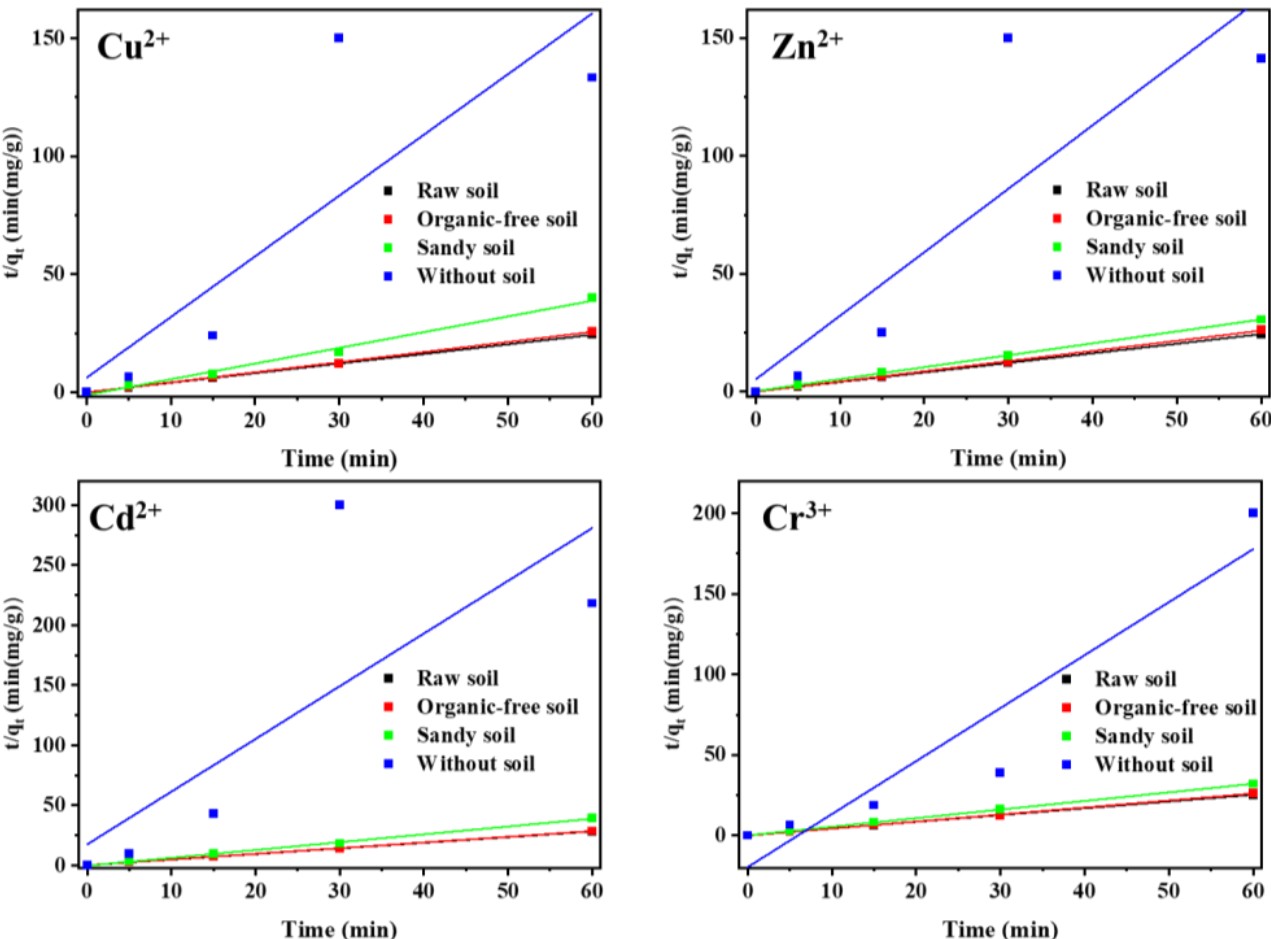

**Figure 4.** The pseudo-second-order kinetics model of trace metal removal by different substrate in the alkali–soil system. The pH of the synthetic leachate was adjusted to 8.5.

Commonly, trace metal cations are adsorbed through chemically bonding with the active site. According to previous research, the functional groups developed on soil act as active centres and form complex structures with metal cations, as shown in Equations (5) and (6) [44,45]:

$$2(\equiv SiO^-) + M^{2+} \rightarrow (\equiv Si - O)_2 M \tag{5}$$

$$2(\equiv AlO^-) + M^{2+} \rightarrow (\equiv Al - O)_2 M \tag{6}$$

Therefore, sandy soils with lower contents of silicate and aluminate show a lower adsorption capacity for trace metals [46,47]. In conclusion, the natural organic matter, silicate and aluminate of soil are the main binding sites for trace metal retention.

### 3.6. Interactions between the Soil and Leachate

Figure 4 shows that inorganic components in soil play a key role, while organic matter has a limited effect on metal ion removal in alkali solution. To explain the removal mechanism of trace metals, FTIR and XPS analyses of the original and soil used with the authentic leachates were performed. Their FTIR spectra in the region of 4000–400 cm$^{-1}$ are displayed in Figure 5. The band at approximately 3440 cm$^{-1}$ is generally assigned to O–H and/or N–H groups [48]. The peaks at approximately 2926, 2354 and 1630 cm$^{-1}$ can be assigned to C–H stretching, $CO_2$, and asymmetric stretching of COO bands. The vibrations of carbonate and quartz give rise to bands at approximately 1425 and 778 cm$^{-1}$,

respectively. The band at 500 cm$^{-1}$ and 1032 cm$^{-1}$ is assigned to Si–O of silicate [49]. After sorption, the soil surface was covered by the metallic hydroxides, and those peaks were weakened and even disappeared. The slight band shift and strength change of the peaks corresponding to O–H/N–H and COO$^-$ during the process of tailing adsorption suggest that there are small quantities of metal ions and organic matter complexes in the soil. There were no other observable differences in the FTIR spectra, indicating that the organic matter in soil is a subordinate factor for metal ion removal.

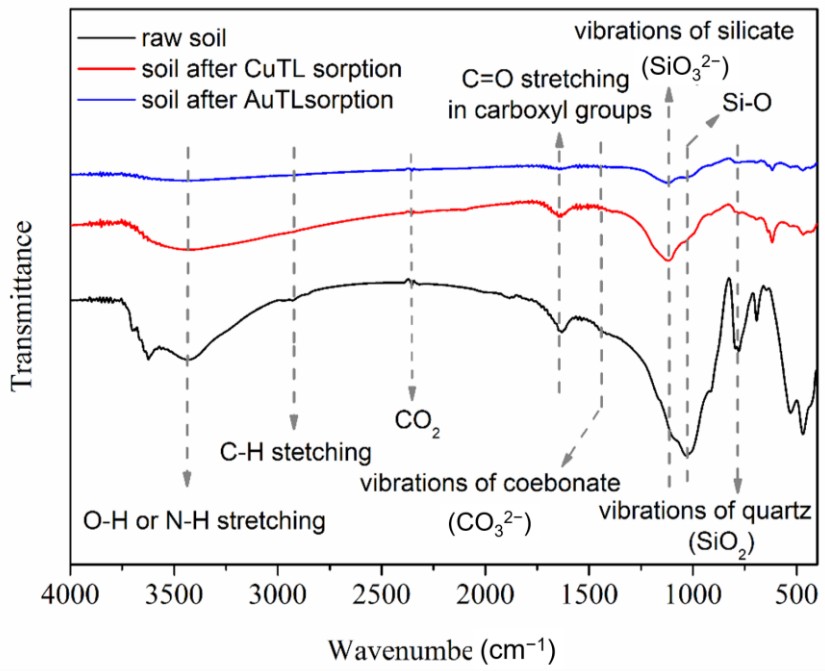

**Figure 5.** The FTIR spectra of raw soil, soil after copper tailing leachate (CuTL) and gold tailing leachate (AuTL).

Wide scan spectra of raw soil and soil after CuTL and AuTL sorption are shown in Figure 6. The peaks at approximately 74, 102, 284, 532, and 713 eV were attributed to Al2p, Si1s, C1s, O1s, and Fe2p, respectively. The C1s spectrum of raw soil comprised five peaks with differentiated binding energy values after deconvolution, which were attributed to sp$^2$ hybridized graphitic carbons (284.4 eV), sp$^3$ hybridized diamond-like carbon (285.0 eV), carbon atom peaks connected to oxygen atoms (C–O, 285.6 eV), C=O (286.7 eV), and COOH (COOR) (289.0 eV) [50,51]. After CuTL and AuTL sorption, the C=O peak disappeared and the binding energies of C–O and COOR/COOH decreased, which is additional evidence of the formation of organic matter–metal complexes.

It can be deduced from the abovementioned discussions that the inorganic matter of soil plays a pivotal role in metal ion removal. The charge property of the soil was determined [52], and the pH$_{PZC}$ was 9.50 (Figure 7a), indicating that the surface of the soil was only slightly charged under the experimental conditions (pH 8.5). Hence, static electricity interactions between metallic hydroxides and soil may not impact the removal of these hydroxides. The adsorption and deposition of metallic hydroxides on the soil surface should thus be the dominant path of metallic hydroxide removal. The XRD patterns of the raw soil and soil after CuTL and AuTL sorption are shown in Figure 7b. Apart from the dominant crystalline phases of SiO$_2$, Al$_2$O$_3$, and MgO, new crystalline phases were not clearly found in the soil after the adsorption of leachates, indicating that metallic hydroxides were deposited and adsorbed on the surface of the soil in amorphous forms. We adjusted the pH of the authentic leachates (CuTL and AuTL) to 8.5 and observed that stable colloidal solutions formed for both leachates (Figure 7c). Both colloids were

completely resolved when the pH was adjusted to be acidic, demonstrating that the colloids are amorphous metallic hydroxides.

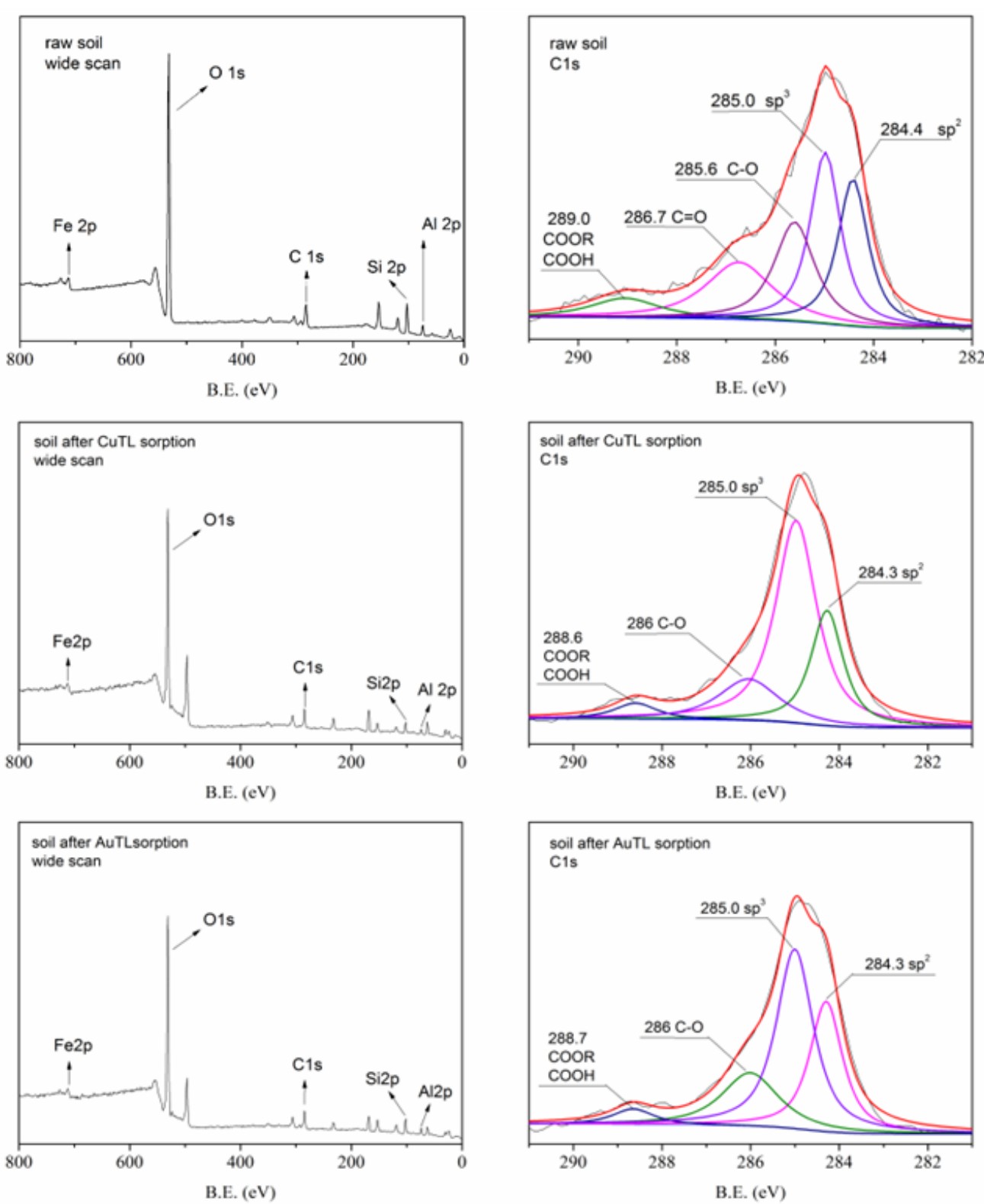

**Figure 6.** The wide scan and C 1s XPS spectra of raw soil, soil after copper tailing leachate (CuTL) and gold tailing leachate (AuTL).

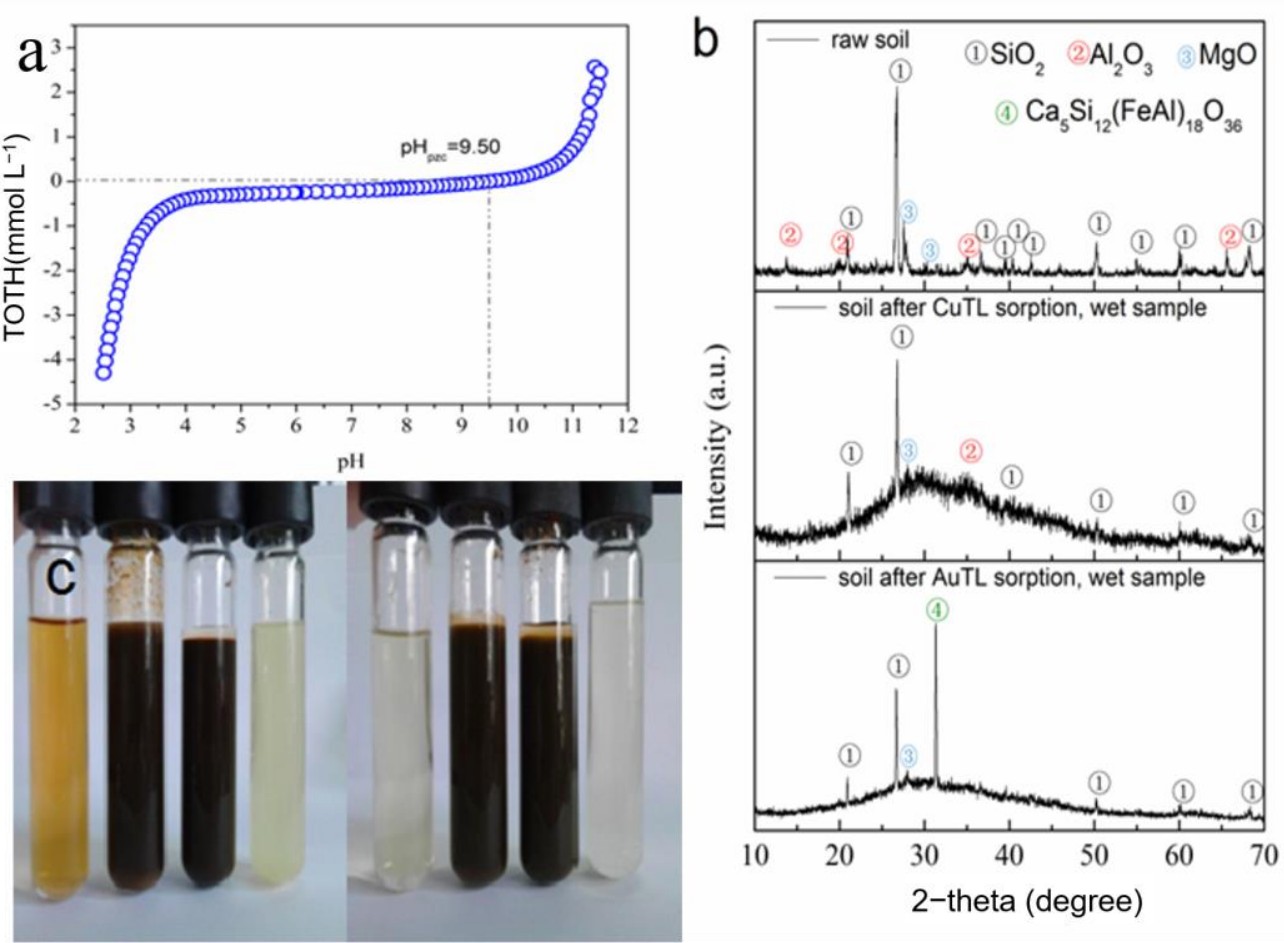

**Figure 7.** (**a**) The pH$_{PZC}$ of soil, (**b**) XRD spectra of original and used soils (wet), and (**c**) photographs of original, alkalized, alkalized after 60 min standing, and acidified leachates (left: CuTL, right: AuTL).

### 3.7. Estimation of the Consumption of Ca(OH)$_2$ in an Alkali–Soil Suspension System

The volume of treatment equipment can be chosen according to the economically optimized HRT. The consumption of Ca(OH)$_2$ for the disposal of mine tailing leachate is mainly governed by two factors. On the one hand, the pH of the tailings leachate needs to be adjusted from acidic to alkaline (8.5). If the leachate pH is 5.0, acid–base equilibrium calculations show that approximately 0.49 g of Ca(OH)$_2$ (s) per cubic metre would be required to bring the pH to 8.5. On the other hand, the densities of gold tailing leachate ($\rho$[Au]) and copper tailing leachate ($\rho$[Cu]) were 1074.9 and 1114.9 kg m$^{-3}$, respectively, and the contents of Cu$^{2+}$, Zn$^{2+}$, Cd$^{2+}$, and Cr$^{3+}$ were 7.94, 8.06, 0.04, and 40.54 g ton$^{-1}$ in the gold tailing leachate and 126.02, 124.59, 1.70, and 5.60 g ton$^{-1}$ in the copper tailing leachate, respectively. In the authentic leachate treatment experiments, the removal efficiency of trace metals was almost 100%, and 104.94 g (9.25, 9.12, 0.02, and 86.55 g for Cu$^{2+}$, Zn$^{2+}$, Cd$^{2+}$, and Cr$^{3+}$, respectively) and 300.91 g (146.86, 140.97, 1.12, and 11.96 g for Cu$^{2+}$, Zn$^{2+}$, Cd$^{2+}$, and Cr$^{3+}$, respectively) of Ca(OH)$_2$ were consumed to precipitate these metals in the gold tailing leachate and copper tailing leachate, respectively. The total consumption of Ca(OH)$_2$ for 1.0 t of gold and 1.0 t copper tailing leachate were approximately 105 and 301 g, respectively.

### 3.8. Reclaiming of Trace Metals

It is necessary to dispose of the polluted soil suspension after retaining the trace metals from the tailing leachates. The recovery rates of the four metal ions from the AuTL- and CuTL-polluted soils are presented in Figure 8. Among these four metal ions, the recovery

rate of $Cu^{2+}$ was the highest, at 72.2% and 95.5% for the AuTL- and CuTL-polluted soils, respectively, demonstrating that $Cu^{2+}$ can be reclaimed efficiently by acid. A total of 59.3% of the $Zn^{2+}$ was reclaimed by $H_2SO_4$ from CuTL-polluted soil, while only 30.6% was reclaimed from the AuTL-polluted soil. Nevertheless, the recovery rates of $Cd^{2+}$ and $Cr^{3+}$ were relatively low from both the AuTL- and CuTL-polluted soils due to their very low concentrations.

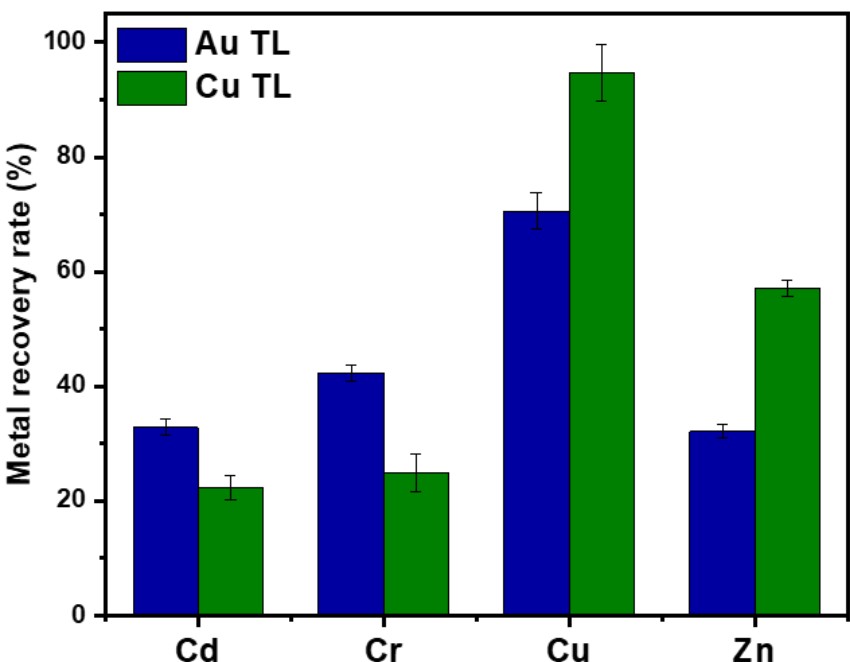

**Figure 8.** The reclaim rate of trace metals in authentic tailing leachates.

### 3.9. LCA Results

Depending on the type of environment affected, impact assessment is currently divided into three main categories, namely resource depletion, ecological impact, and environmental health. Through the CML 2 baseline 2000 v2.04/World, 1990 evaluation method, impacts are further divided into abiotic depletion, acidification, eutrophication, global warming (GWP 100), ozone layer depletion, human toxicity, freshwater aquatic ecotoxicity, marine aquatic ecotoxicity, terrestrial ecotoxicity, and photochemical oxidation. Table 3 shows the impact of the assessment phase results. The operation system accounts for most of the total score in all categories and has the greatest impact on the environment, which is mainly due to the high electricity consumption during the reaction process [53]. The addition of chemical reagents to the tailing leachates greatly affected the environment, including ozone layer depletion (33.8%), acidification (15.1), eutrophication (14.9), and photochemical oxidation (14.8), which was mainly caused by the addition of NaOH to neutralize acid wastewater. Waste handling systems mainly influence freshwater aquatic ecotoxicity (10.8%) due to the residual metals in wastewater. Overall, marine aquatic ecotoxicity is the most significant impact during tailings leachate remediation, followed by GWP 100 and human toxicity. The total GWP 100 of the alkali suspension method was 1.353231 kg of $CO_2$ eq, which was lower than the 1.86 kg of $CO_2$ eq with the traditional lime-based treatment process [54], indicating a lower environmental impact on climate change.

**Table 3.** Impact results for the project by characterization.

| Impact Category | Unit | Chemical Reagents | Waste Handling | Transportation Energy | Separation Energy | Operation Energy | Total |
|---|---|---|---|---|---|---|---|
| Abiotic depletion | kg Sb eq | 0.00064 | $1.67 \times 10^{-5}$ | $4.8 \times 10^{-5}$ | 0.00017 | 0.00973 | 0.010605 |
| Acidification | kg $SO_2$ eq | 0.001256 | 0 | $2.63 \times 10^{-5}$ | $9.09 \times 10^{-5}$ | 0.006928 | 0.008301 |
| Eutrophication | kg $PO_4$ eq | $7.74 \times 10^{-5}$ | 0 | $3.3 \times 10^{-6}$ | $7.23 \times 10^{-6}$ | 0.000445 | 0.000533 |
| Global warming | kg $CO_2$ eq | 0.094831 | 0 | 0.007982 | 0.02362 | 1.226799 | 1.353231 |
| Ozone layer depletion | kg CFC-11 eq | $1.74 \times 10^{-8}$ | 0 | $6.35 \times 10^{-9}$ | $1.5 \times 10^{-9}$ | $2.76 \times 10^{-8}$ | $5.28 \times 10^{-8}$ |
| Human toxicity | kg 1,4-DB eq | 0.007139 | $8.41 \times 10^{-5}$ | 0.001576 | 0.006908 | 0.247261 | 0.262968 |
| Fresh water aquatic ecotoxicity | kg 1,4-DB eq | 0.001418 | 0.006579 | 0.000114 | 0.001515 | 0.049796 | 0.059422 |
| Marine aquatic ecotoxicity | kg 1,4-DB eq | 2.9444 | 1.017661 | 0.270359 | 4.455857 | 124.7827 | 133.471 |
| Terrestrial ecotoxicity | kg 1,4-DB eq | $5.7 \times 10^{-5}$ | $2.83 \times 10^{-25}$ | $3.28 \times 10^{-6}$ | $4.08 \times 10^{-5}$ | 0.005705 | 0.005806 |
| Photochemical oxidation | kg $C_2H_4$ | $4.13 \times 10^{-5}$ | 0 | $4.48 \times 10^{-6}$ | $3.69 \times 10^{-6}$ | 0.000252 | 0.000302 |

## 4. Conclusions

In this study, an alkali suspension method was proposed to treat mine tailing leachates, and the reaction parameters for the removal of trace metals were optimized in synthetic and authentic tailing leachates. Under optimized conditions, with a buffer pond pH of 8.5, HRT of 5 min, and soil amount of 20 g L$^{-1}$, most of the trace metal can be removed from solution with a removal rate near 100%, demonstrating that this method is a feasible method to efficiently prevent the risks of tailing ponds caused by leakage or dam breakage. Based on the characterizations and control experiments, a removal mechanism was proposed by which the inorganic matter in soil can remove amorphous metallic hydroxides by deposition and adsorption, and the organic matter in soil can form metal–organic complexes to remove trace metals. In addition, the theoretical consumption of $Ca(OH)_2$ for 1.0 t of gold and 1.0 t of copper tailing leachates was calculated to be 105 and 301 g, respectively. Notably, more than 90% of the $Cu^{2+}$ in authentic Cu tailing leachate can be reclaimed using this method.

**Supplementary Materials:** The following supporting information can be downloaded at: https://www.mdpi.com/article/10.3390/w15101902/s1, Figure S1: The flow chart of the alkali suspension method; Figure S2: General system boundaries for tailing leachates treatment though alkali suspension method; Figure S3: Effect of different soil concentrations on the $Cu^{2+}$ during the reaction; Figure S4: Concentration-time curves of different trace metal; Figure S5: The pseudo-first-order kinetics model of trace metal removal in the alkali-soil system; Figure S6: Removal efficiencies of trace metals in solutions treated with raw soil, organic-free soil, sandy soil, and without soil; Table S1: Initial concentrations of Cu, Zn, Cd and Cr in authentic copper and gold tailing leachates; Table S2: Theoretical equilibrium concentrations of different trace metals at pH of 8.5, which were calculated based on solubility product constant ($K_{sp}$).

**Author Contributions:** S.J.: Conceptualization, Supervision, Writing—Original draft preparation. Y.C.: Formal analysis, Data Curation. S.C.: Investigation, Methodology. Z.H.: Visualization, Writing—Original draft preparation. All authors have read and agreed to the published version of the manuscript.

**Funding:** The authors gratefully acknowledge financial support from National Natural Science Foundation of China (21876166).

**Data Availability Statement:** The datasets used and/or analysed during the current study are available from the corresponding author on reasonable request.

**Conflicts of Interest:** The authors declare no conflict of interest.

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
