# Peer review of "Removal and Reclamation of Trace Metals from Copper and Gold Mine Tailing Leachates Using an Alkali Suspension Method"

_water, doi:10.3390/w15101902_

Round 1

Reviewer 1 Report

The authors developed an efficient and sustainable method for the capture of trace metals from tailing wastewaters The work is well written and needs some corrections before being considered for publication.

1.       Adsorption techniques should be mentioned in the introduction, as they prove to be overall sustainable owing to the possibility of biomass-derived adsorbents and continuous reuse of the materials.

2.       Please provide characterisation information of the real leachates in the methodology part.

3.       Check the spelling of the hydroxide anion. The negative should be near the O, not H.

4.       The speciation of copper seems wrong. There is no copper at pH 8. Also correct the y axis name.

5.       Could the removals observed in Figure 2 be attributed to precipitation which hindered the real determination of metal concentration?

6.       What is the conclusion between the different temperatures?

7.       Please provide the qe values for the metals.

8.       The authors must provide tests of different soil concentrations.

9.       Figure 3a is confusing, the values should be given in a table or by text. What is the error of these measurements?

10.   Soil removes nearly 100% of all metals at pH 8.5, why not skip pH adjustment and use the raw leachate? Why does Cd removal decrease at 30C?

11.   Why do the IR peaks near 500 cm-1 disappear after metal sorption? Correct “coebonate” and “stetching”.

12.   The PZC can be given simply by the third graph of Figure 7a.

13.   Triplicates (errors) need to be provided for the metal reclamation experiments.

English is fine.

Reviewer 2 Report

The authors present proposed a quick and cost-effective soil-assisted alkali suspension approach to remove and reclaim the trace metals. The removal efficiencies fluctuated slightly with temperature and the organic matter content in soil. More than 90% of the Cu2+ in authentic Cu tailing leachate can be reclaimed by simple acid treatment. This study provides an efficient and low-cost approach for the disposal and recycling of toxic mine tailing leachates. Although this work is interesting, I feel that there are some places that need in-depth study. So, the paper needs minor revision before acceptance for publication. My detailed comments are as follows:

1. Introduction needs a detailed review of previous work; the author can refer to these papers. What are the advantages of method compared with other treatment technologies, and where are the characteristics and advantages of the method in this paper?

2. If possible, the experimental data needs to be analyzed with error bars.

3.The author can include the experimental section Fig. S2-4 in the main text, as this data is very important.

4. If possible, electron microscopy SEM or TEM can be used to analyze the morphology and composition of the soil (for example, Chem. Eng. J. 409, 2021, 128185).

OK

Reviewer 3 Report

1. Introduction section is weak, add more recent papers 

2. Add Methodology flowchart in the present study

3. Add the flow chart to clear about the methodology 

4. Add more reference related to this study and compare the results, 

5. Revise the conclusion based on the correction carried out in the main text

Minor revision 

Reviewer 4 Report

The manuscript (MS) proposed by Shun-Feng Jiang et al. “Removal and reclamation of trace metals from copper and gold mine tailing leachates using an alkali suspension method” is a suitable topic; however, analyzing the results sent in the form of tables and figures, I could see that the authors did not present any standard deviation of the data. This is very precarious, given that since it is a scientific study, it seems to me that there is a great lack of scientific rigor. Thus, the manuscript can be accepted after major revision. 

Point 1: Abstract. This section is critical for others to read the paper, thus, the authors should improve the abstract. (more attractive).

Point 2: Figure 1. What is the meaning of ‘mol. %’?

Point 3: Page 6 line 225. The authors confound the term ‘correlation coefficient’ (R) with the term ‘coefficient of determination’ (R2). The R is sometimes criticized as having no obvious intrinsic interpretation, and researchers sometimes report the square of the R, that is, the R2. It can be interpreted as the proportion of variance in 1 variable that is accounted for by the other. (Am Stat. 1988;42:59–66; Anesth Analg 2018;126:1763).

Point 4: Do not use or define acronyms if they are not reused in the research highlights; define acronyms when first used in the highlights, if used more than once.

- Text: Define acronyms when first used in the text and only once.

- Text: Do not define acronyms if they are not used in the text.

- Text: Spell out acronyms that are not reused in the text, captions, or defined.

- Note that the abstract and the text are two separate entities. 

- Abstract: Define acronyms when first used in the abstract (if they are used in the abstract).

- Abstract: Do not define acronyms if they are not used in the abstract. 

- Abstract: Spell out acronyms that are not reused in the abstract nor defined.

Point 5: Minor remarks: 

- Improve all the figure captions.

- Remove equation 3, unnecessary.

Very good.

Round 2

Reviewer 1 Report

The authors have answered all concerns. The manuscript is able to be published in its current format.

The English could be improved throughout the text. Many instances of wrong subject/verb and incorrectly placed adjectives.

Reviewer 4 Report

Accept in present form.

Good.